# Impact of High-Speed Rail on Spatial Structure in Prefecture-Level Cities: Evidence from the Central Plains Urban Agglomeration, China

**Xiaomin Wang** [1], **Jingyu Liu** [1,*] and **Wenxin Zhang** [2]

1   College of Geography and Environmental Science, Research Center of Regional Development and Planning, Henan University, Kaifeng 475004, China
2   Faculty of Geographical Science, Beijing Normal University, Beijing 100875, China
*   Correspondence: 10130021@vip.henu.edu.cn

**Abstract:** The impact of high-speed rail (HSR) on urban spatial structure has attracted much attention since the 1970s. It mainly realizes the change of urban spatial structure by affecting the spatial distribution of population and economy. Based on population and industry data on 29 cities in the Central Plains Urban Agglomeration (CPUA) located in central China during 2005–2017, in this paper difference-in-difference (DID) models are utilized to explore the influence of HSR on the spatial structure of prefecture-level cities at the single/multicenter levels according to the dimensions of polycentricity and dispersion. The main conclusions are shown as: (1) HSR on the spatial structure of the CPUA has a monocentric trend, but the characteristics of different prefecture-level cities vary. Currently, agglomeration remains the dominant force of the spatial structure of the CPUA in prefecture-level cities. (2) HSR have a noticeable effect on the spatial structure of industry, whereas the influences of the HSR on the spatial structure of the population are insignificant. Its development is the result of the combined effects of many factors, including natural factors, socioeconomic factors, policies, and transportation factors. Among the controlling factors, the administrative area, economic development level, urban population, and number of research units are the critical factors having a hold on the population spatial structure of the CPUA.

**Keywords:** spatial structure; population and industry; high-speed rail (HSR); Central Plains Urban Agglomeration (CPUA)

## 1. Introduction

Transportation infrastructure is an important element of human production and life, a means of facilitating interrelationships between different regions, and an essential part of modern civilization. Because of its strong transport capacity, high-speed rail (HSR) is considered to be a significant travel mode and a critical component of transport infrastructure. HSR enables fast delivery, safety, reliability, punctuality, reduced land and energy use, reduced pollution, low accident rates, reduced traffic congestion, and other advantages [1–3]. HSR impacts urban spaces by altering accessibility [4–6]. HSR networks and stations are important carriers of urban interaction, and their construction improves city accessibility [7], especially in station areas. The resulting concentration and diffusion of resources and factors in space profoundly impact urban development and its spatial structure [8–15].

HSR brings time-space compression effects to cities [16,17]. If a region has good accessibility coupled with a high proportion of information exchange industry and ample opportunities for higher education, HSR can improve urban accessibility, promote population mobility, create many employment opportunities, and drive population growth in cities along the lines [18–20]. Cities with stations along HSR lines have 16–34% greater employment in business, wholesale and retail, catering, and accommodation services than

those in cities without HSR lines, bringing 15–20% more economic growth to development zones [21–23]. The population growth rate of "Shinkansen" HSR line in Japanese cities is 22% higher than that of cities not along HSR lines [24,25].

In short, the development of HSR networks may provide greater economic and employment growth for core cities and further increase the differences between core (with their own HSR network) and marginal cities (intermediate cities and low-density regions without an HSR) [24–28]. Economic development tends to expand and converge along transportation axes such as national highways and railway lines, changing the external shapes of cities and showing obvious traffic directionality. Good traffic conditions can form a competitive location advantage and attract population and industry clusters [29,30]. HSR can change regional location conditions and impact the redevelopment of urban space, which can strengthen the extension of the original center and provoke the growth of a multicenter. There are also many studies on whether HSR leads to regional spatial dispersion or concentration. Some studies show that HSR will strengthen regional integration and form corridor economy, and regional HSR will lead to more concentrated spatial pattern [3,17].

While improving urban accessibility, new HSR stations and the renovation of existing stations have successfully attracted urban activities such as business, office and residence, enriched urban functions, and then boosted the formation and reconstruct of urban spatial structure [31]. The impact of HSR on urban development includes both the development of station areas and the construction and updating of many urban projects; thus, HSR serves an essential mechanism in the overall spatial development of cities, counties, and county-level cities in the station areas and in guiding the city [32–34]. The redistribution of spatial resources has disrupted the original urban spatial structure and restructured it.

Most studies of the impact of HSR have focused on accessibility [35], geographical discrepancy [3,36], spatial development, economic development, and commuting behaviors [37]. Although some studies have examined the effects of HSR on the spatial structure of the region and stations, the impact of HSR on spatial structure in prefecture-level cities (which are ranked third in the Chinese administration hierarchy, implementing the administrative system of City-Governing-County system) remains unexplored. At present, most studies of the polycentricity of urban agglomerations have explored the morphological dimension using population data for analysis and lack a multidimensional research perspective. The purpose of this study is to utilize traditional and continuous difference-in-difference (DID) models to explore the impact of HSR on urban spatial structure in prefecture-level cities based on two perspectives of population and industry. The structure of this article is as follows: Section 2 presents the current studies analyzing the urban spatial structure impacts of HSR. Section 3 presents overview of the study area and on related research methods. Section 4 is benchmark regression analysis. Section 5 provides mechanism analysis. Section 6 presents conclusions and a discussion.

## 2. Literature Review

HSR affects urban space directly by improving accessibility [38–40]. Accessibility is defined as the ability of nodes to interact with each other [41]. Advances in transportation networks and relevant research have continually enriched and developed the connotation of accessibility. Accessibility is reflected in the convenience of human activities involving movement from one region to another by means of the transportation network, which can be measured by travel time, cost, etc. [42]. Indicators of HSR accessibility include the weighted average travel time, contour measure, spatial separation measure, and network density [6]. Most research shows that HSR construction greatly shortens intercity travel time, brings obvious "space-time convergence" to the region, and greatly optimizes the overall location condition of a city and its position in the city network [40,43].

HSR changes the location accessibility of different nodes in the region [44]. HSR can provide services for 74% of population and 82% of GDP in a two-hour travel time. The provision and scale of HSR have stirred up business service and land market and boosted the economy expansion by 14% [8]. Through the imposition and weakening of different

nodes, HSR objectively contributes to the redistribution of economic activities by attracting external personnel, materials, information, and other factors [18,45,46]. New investment opportunities lead to changes in the structure of land use and space and the formation of clusters of different levels, which become constituent elements of cyberspace in cities or larger areas [47]. HSR shortens the space-time between regions, increases the possibility of intercity connection, leads to the change of urban location conditions, reallocates various resource elements in the area, integrates the economy, reshapes the urban system, and rebuilds the spatial structure.

HSR mainly impacts urban spatial structure by affecting the spatial distribution of the population and economy [34]. HSR strengthens existing urban centers and promotes the creation of new urban centers. HSR stations in different locations have different spatial impacts. HSR stations in the central areas of cities are updated based on the original stations when they are very convenient for connecting with the subway and public transportation in the city, which helps to strengthen the original urban center. Due to the "catalyst effects" of HSR hubs, capital, industry, and technology continue to accumulate in the HSR station area as the core urban space, and this station area becomes an important growth center leading the development of the region. Substantial integration of HSR and the urban internal transportation network with the direction of urban planning can promote the formation of urban polycentric spatial structures [20,32,33,48]. Policy has a significant impact on urban spatial expansion [49]. When HSR enters the urban system as a new factor, it needs to be coordinated with and adapted to the urban spatial development pattern and trends to deliver the maximum effect. HSR station locations that are consistent with the established and planned direction of the urban environment will be instrumental in rational collocation of the city, multiple essential factors of station, and the cultivation of a new growth pole.

The establishment of an HSR hub has a catalyst effect on the development of the surrounding space, triggering new development and transformation in the area and driving changes in the urban spatial structure. The varying space-time accessibility around HSR stations will lead to differences in the concentration of various factors and the formation of different industrial agglomeration and impact zones [50]. With the HSR hub as the center point, the station area can be divided into a core area, an expansion area, and an impact area, forming a circular spatial structure [51]. Schutz E, POL, et al. applied the circle structure model theory of the "three development zones" to the development of HSR stations in different regions to distinguish three circular zones [52,53]. This model divides the periphery of the transportation hub area into three circular zones. The first, second, and third development zones represent the range of stations that can be reached within ten minutes, and the functional characteristics of each of the different circles have been studied. Scholars have conducted in-depth studies in different regions and at different sites to modify and improve Schutz's three development zones circle structure theory [54,55].

## 3. Settings, Data, and Method

### 3.1. Research Area: The CPUA

The Central Plains Urban Agglomeration (CPUA) as the representative of underdeveloped regions is located in central China; it is located in the heart of China. The "meter" shaped transportation network connects the north and the south and connects the east and the west. Therefore, based on the analysis of the impact of HSR on the urban spatial structure of underdeveloped areas, it provides theoretical support and guidance for the optimization of the spatial structure of underdeveloped areas. It is strategically positioned as an important comprehensive transportation hub. According to "The CPUA planning outline", the CPUA encompasses Henan, Shanxi, Hebei, Shandong, and Anhui Provinces, covering 30 cities and 3 counties. It includes 18 cities in Henan Province, as well as the surrounding Jincheng, Changzhi, Yuncheng, Handan, Xingtai, Liaocheng, Heze, Huaibei, Bozhou, Suzhou, Fuyang, Bengbu, and Huainan Panji districts and Fengtai and Dongping counties. This study mainly examines the impact of HSR on the spatial structure of prefecture-level cities in the CPUA. Therefore, Jiyuan and county-level cities are excluded,

and only 29 prefecture-level cities are considered (excluding Jiyuan because Jiyuan is a county-level administrative unit directly under the provincial government).

With the complexity of the evolution of the transport infrastructure, the traffic infrastructure in the CPUA has experienced three stages: the highway stage; the conventional rail stage; and the HSR stage. In 2010, the Zhengzhou-Xian HSR, the first HSR in the CPUA, opened the era of new HSR in the CPUA. Subsequently, the CPUA entered an HSR expansion period, with the successive opening of the Beijing-Shanghai HSR, Shijiazhuang-Wuhan HSR, Zhengzhou-Kaifeng intercity HSR, Zhengzhou-Jiaozhou intercity HSR, and Zhengzhou-Xuzhou HSR. In 2017, the HSR network formed a "cross" type (Figure 1). The "meter" HSR network with Zhengzhou as the core was constructed in 2020. In this study, the total population and GDP of municipal-level cities were used to calculate the spatial structure index. The urban spatial structure in the CPUA 5 years before HSR operation was compared with that 7 years after the operation of HSR in prefecture-level cities.

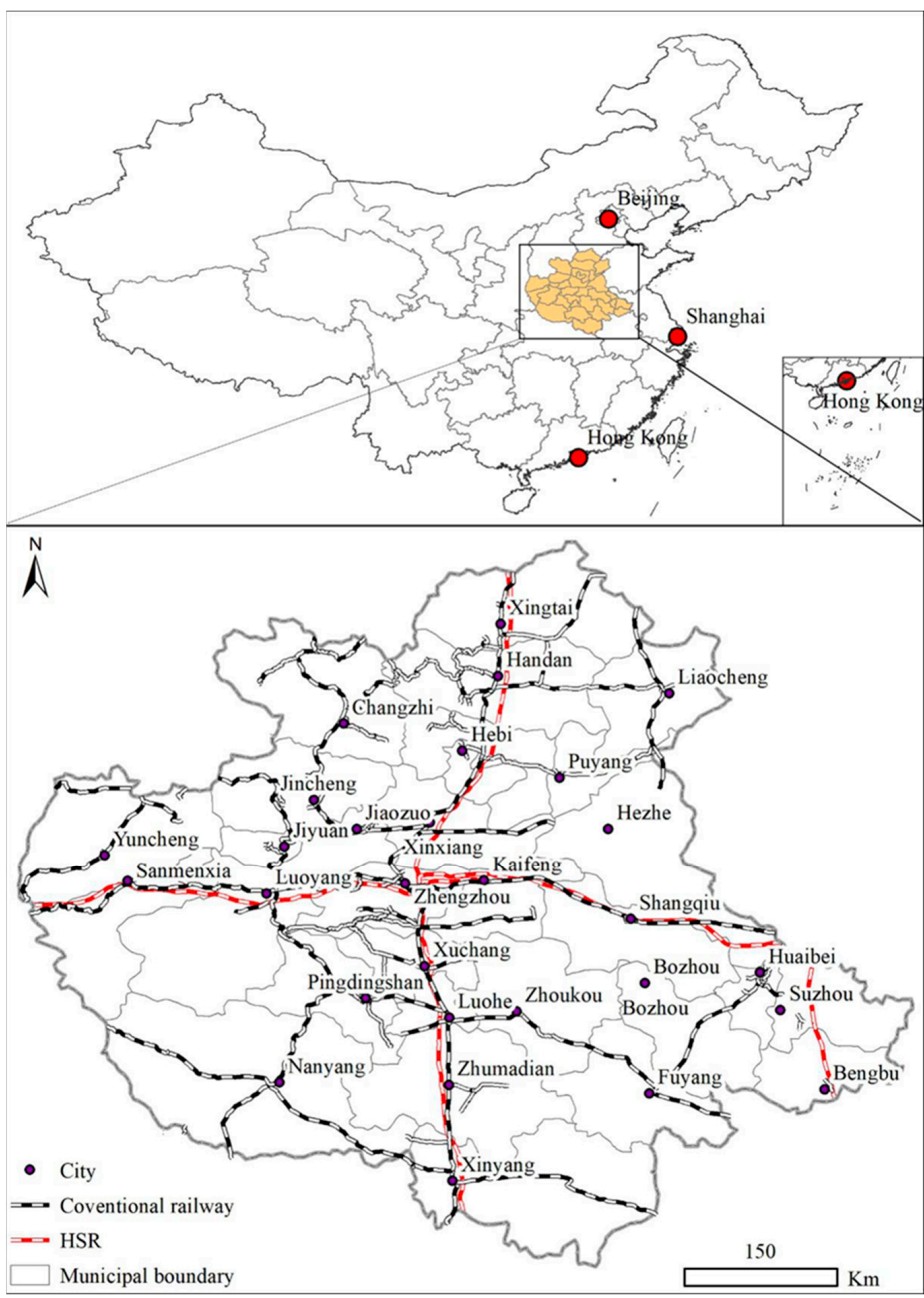

**Figure 1.** Railway network in the CPUA of China.

### 3.2. Data Description

The data on the HSR network and stations in the study area are mainly from the map of China's 1:4 million highway transportation edition published by the Ministry of Communications of China and the 2017 operation route map of China's HSR. HSR frequency data are obtained from the 2010–2017 railway timetable posted by the Ministry of Railways of China. The resident population and GDP data for calculating urban spatial structure indicators and control variables are from the 2006 to 2018 provincial statistical yearbooks. Table 1 shows the descriptive statistics of various variables.

**Table 1.** Definitions and descriptive statistics of variables.

| Variable | Mean | Standard Deviation | Minimum | Maximum |
|---|---|---|---|---|
| HSR | 0.66 | 0.48 | 0 | 1 |
| GDP | 6,101,990 | 7,304,461 | 278,624 | 67,000,000 |
| Resident population | 209 | 117 | 55 | 714 |
| Administrative area | 9804 | 5165 | 2160 | 26,509 |
| The per capita GDP | 26,077 | 14,599 | 3761 | 93,792 |
| Proportion of secondary and tertiary industry | 84.58 | 8.10 | 59.47 | 98.35 |
| Proportion of government expenditure | 47.53 | 63.80 | 12.65 | 1078.10 |
| County-level unit | 6.74 | 3.63 | 1.00 | 17.00 |

### 3.3. Methodology

3.3.1. Measurement Method for Spatial Structure

The rule of rank size, the Herfindahl index, and the Primacy index are commonly applied for the sake of estimate the monocentric and polycentric spatial structure of a population. In this study, the Herfindahl index was selected to estimate the extent of monocentric and polycentric structure of the population of CPUA prefecture-level cities.

The Herfindahl index is as follows:

$$HHI = \sum_n^i \left( \frac{S_i}{S} \right)^2 \tag{1}$$

where $S_i$ represents the population size of unit $i$ within the city and $S$ represents the population size of the entire city. The value of the Herfindahl index is, and an index value closer to 1 (0) indicates that the spatial structure of the population tends to be monocentric (polycentric).

The spatial Gini coefficient was selected with the purpose of estimating the degree of the concentration ratio of the spatial structure of industry in CPUA prefecture-level cities in this study. The spatial Gini coefficient takes the industrial share as a substitution variable to calculate the regional industrial agglomeration. It does not consider the population scale and economic scale of the region for the sake of decreasing the affect of the substitution variable on spatial structure index. The formula of the spatial Gini coefficient is as follows:

$$G = \frac{1}{2T(n-1)} \sum_i \sum_j |P_i - P_j| \tag{2}$$

where $T$ represents the total industry of $n$ units in the city, and $P_i$ represents the industry scale of unit $i$ within the city. The value of $G$ is. The larger the $G$ is, the greater the gap in industrial shares between regions, the obvious industrial advantages in a few regions, and the higher the degree of industrial agglomeration. In contrast, the smaller the spatial Gini coefficient is, the lower the level of industrial agglomeration.

### 3.3.2. Model Specification

Recently, many academics have paid close attention to the causal relationships between transportation infrastructure and urban environments. The main research methods include structural equation modeling and DID modeling [18,49,56]. The core idea of DID is random experiment as a causality effect. The idea of this model is to divide the samples into an

experimental group and a control group and then analyze the changes of the two groups of samples after the introduction of policies. The impact of HSR on urban spatial structure is primarily showed into two terms: the "Time Effect" and the "HSR Effect". To better study urban development caused by HSR, the time effect could be separated. Therefore, we follow Wooldridge's quasi-natural experiment and use the DID model to measure the HSR effect. The baseline regression model is as follows:

$$y_{it} = \alpha_0 + \beta_0 time_{it} * city_{it} + \gamma Z_{it} + \delta_{it} + f_{it} + \varepsilon_{it} \tag{3}$$

where variable $y_{it}$ represents the spatial structure in the prefecture-level city; $time_{it}$ is a time dummy variable that equals 0 before the HSR opens and 1 after the HSR opens; $city_{it}$ is a prefecture-level city dummy variable that equals 1 if the prefecture-level city has an HSR network and 0 otherwise; $\beta_0$ represents the net impact of HSR on the spatial structure; $Z_{it}$ represents control variables; $\gamma$ is the coefficient of $Z_{it}$; $\delta_t$ is the time fixed effect; $f_i$ is the city fixed effect used; and $\varepsilon_{it}$ is a random interference term.

If we set HSR and non-HSR cities as the experimental and control groups, respectively, the differences in spatial structures between the two groups could be considered to only have been caused by HSR. DID is used to test the parallel trend hypothesis. There are two common test methods: the first compares the time trends of the means of the dependent variable between two groups and the second investigates the interaction term between the experimental group and time dummy variables before the performance of the policy. We use these two methods to test whether the urban population and industrial spatial structure indexes in the prefecture-level cities of two groups meet the parallel trend assumption. The model for the second method is set as follows:

$$y_{it} = \alpha'_0 + \sum_{t=2007-2009} \beta_t + Experiment_i preyears_i + \beta'_0 time * cityR_{it} + \gamma' Z_{it} + \delta'_i + f'_i + \varepsilon'_i \tag{4}$$

where "Experiment" is the experimental group; preyears is the time dummy variable before HSR operation; and $\beta_t$ is the interaction term.

The interaction term coefficient reflects the distinction between the presence and absence of policies but does not reflect the effect of the degree of policy. In 2017, 19 cities in the CPUA had HSR in operation, but the frequency of services varied greatly. In 2017, Zhengzhou provided more than 490 HSR service trains, whereas Zhumadian, Xingtai, Handan, and Yuncheng provided fewer than 20 HSR service trains. Therefore, we add HSR service differences to the model and establish a continuous DID model. We use continuous variables instead of grouping dummy variables:

$$y_{it} = \alpha''_0 + \beta''_0 Nums_{it} yeary_{it} + \gamma'' Z_{it} + \delta''_i + f''_i + \varepsilon''_i \tag{5}$$

where the variable "Num" is the number of HSR frequency and $\beta''_0$ represents the impact of HSR service. In this study, the following control variables were selected:

(1) The administrative area (adm) measures the natural endowment of an area. Traditional location theory and economic geography affirm the role of geographical conditions in economic agglomeration and the formation and development of cities. According to these theories, natural endowments constitute the initial advantages of a region, and relatively superior natural conditions make it easier to attract people. The area of an administrative region is one of the most important indicators for measuring its carrying capacity. Now that the restrict of the range radiation distance of the growth pole, the larger the area of a region is, the easier it is to form a relatively balanced polycentric spatial structure.

(2) The GDP per capita (peGDP) measures the level of socioeconomic development. According to Friedmann's theory of regional spatial structure, as the economic level improves, the functional relations of urban agglomerations will first become monocentric and then polycentric.

(3)   Differences in industrial structure will cause changes in spatial structure. Faced with high land rents and high wages, standardized mature manufacturing industries have a disposition to transfer to medium-sized and small cities with lower costs. The service industry is highly non-tradable, and its production and consumption are often difficult to separate. The high-end service industry has a higher demand for "face-to-face" communication. Therefore, the service industry prefers large cities with large market potential. According to Fujita and others, the secondary industry of standardized production tends to be decentralized, and the tertiary industry tends to be centralized. Studies have shown that the impact of HSR on industry is primarily concentrated in tertiary industries such as business services, tourism, real estate, and financial services, which can increase regional economic income. Therefore, the proportion of secondary and tertiary industries (P sec-ter ind) is selected to measure the industrial structure.

(4)   The urban population (urbp) measures the urbanization process of an area. The processes of urbanization and spatial structure are both influential and change simultaneously [57]. The development of urbanization provides the fundamental driving force for the industrial spatial layout and has an impact on the spatial structure. We consider the economic development level and the nonlinear relationship between population size and spatial structure by adding the second terms of these parameters.

(5)   The proportion of government expenditures (p gov) represents the dimension of government interpose in the market. If the government tends to support the development of core cities, it may implement policies such as local industrial development policies and preferential tax policies that will make it easier to form a monocentric center and agglomeration spatial structure. If the government pays attention to regional spatial coordination and equity, it will promote coordinated regional development, which may lead to the development of a polycentric, decentralized spatial structure.

(6)   The number of counties/county-level cities (cou) measures the degree of competition between governments at the same level. According to the administrative-level unit, a site can be divided into two categories: the municipal site and the county site. The spatial layout of HSR stations in the CPUA includes station layouts in municipal areas and city layouts in county and county-level cities such as Gongyi South Station (Zhengzhou), Lingbao West Station (Sanmenxia), and Yongcheng North Station (Shangqiu). We investigate the development status of all of the stations and find great variation in the development and construction of the HSR stations in the CPUA. The impact of the development of these stations on the population, employment, intercity travel, and urban spatial structure remain to be explored.

## 4. Results

### 4.1. Analysis of the Measurement Results of Spatial Structure in Prefecture-Level Cities

4.1.1. Analysis of the Spatial Structure of the Population of Prefecture-Level Cities

The Herfindahl index of 29 prefecture-level cities was calculated for 2005–2017; monocentric and polycentric spatial structure values and average values of the populations were obtained. We constructed scatter plots of the spatial structure of the population (Figure 2). In general, the average Herfindahl index continues to rise, and the spatial structure of the population tends to be monocentric. The Herfindahl indexes of 18 prefecture-level cities including Zhengzhou, Kaifeng, Luoyang, Xinxiang, Jiaozuo, Sanmenxia, Xuchang, Nanyang, Xingtai, Handan, and Changzhi have the most obvious upward trend. The Herfindahl indexes of some prefecture-level cities such as Pingdingshan, Anyang, Hebi, Xinxiang, and Zhoukou are relatively stable. The spatial structure of the population of some prefecture-level cities such as Luohe, Shangqiu, and Xinyang shows a weak polycentricity trend. These prefecture-level cities have low Herfindahl indexes ranging from 0 to 0.2, indicating that the internal units of their cities are relatively average. The formation of regional agglomeration has an evenly dispersed polycentric structure.

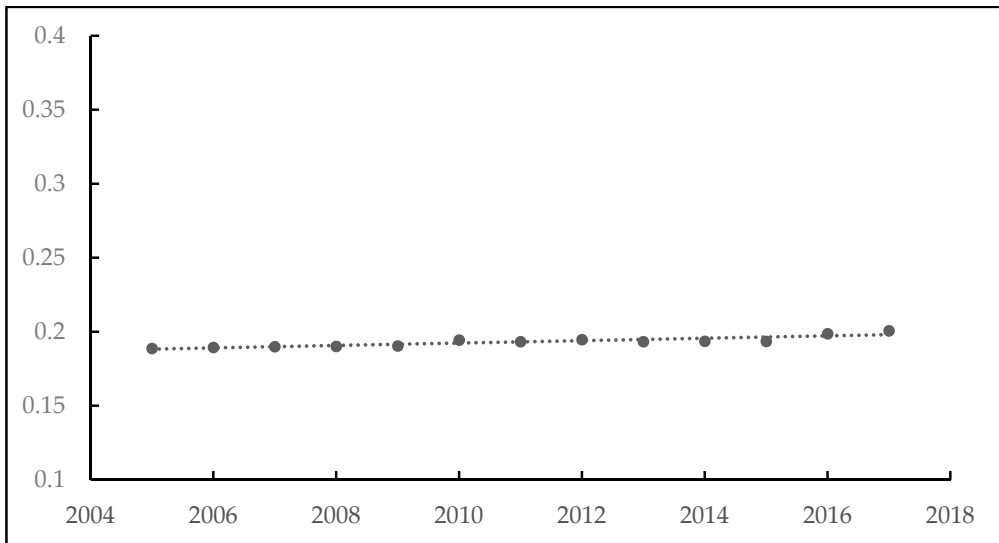

**Figure 2.** Change in the average value of the spatial structure of the population in the CPUA of China.

### 4.1.2. Analysis of the Spatial Structure of Industry in Prefecture-Level Cities

The spatial Gini coefficient index of 29 prefecture-level cities was calculated for 2005–2017, and the spatial structure values and average values of industry were obtained. We constructed scatter plots of the average value of the spatial structure of industry (Figure 3). In general, the spatial Gini coefficient index of the CPUA shows an upward trend, and the overall industrial spatial structure tends toward agglomeration. Thirteen prefecture-level cities show the most obvious upward trend in the spatial Gini coefficient index including Zhengzhou, Kaifeng, Luoyang, Xinxiang, Jiaozuo, Sanmenxia, Xuchang, Nanyang, Xingtai, and Changzhi. Of these cities, 11 are HSR cities, and 2 are not HSR cities, indicating that urban industry is concentrated toward the center in these cities. There are 16 prefecture-level cities in which the spatial Gini index generally shows a downward trend including Handan, Jincheng, Yuncheng, Huaibei, Suzhou, Luoyang, Pingdingshan, Nanyang, Hebi, Liyang, and Luohe, indicating that urban industry is trending toward the periphery in these cities.

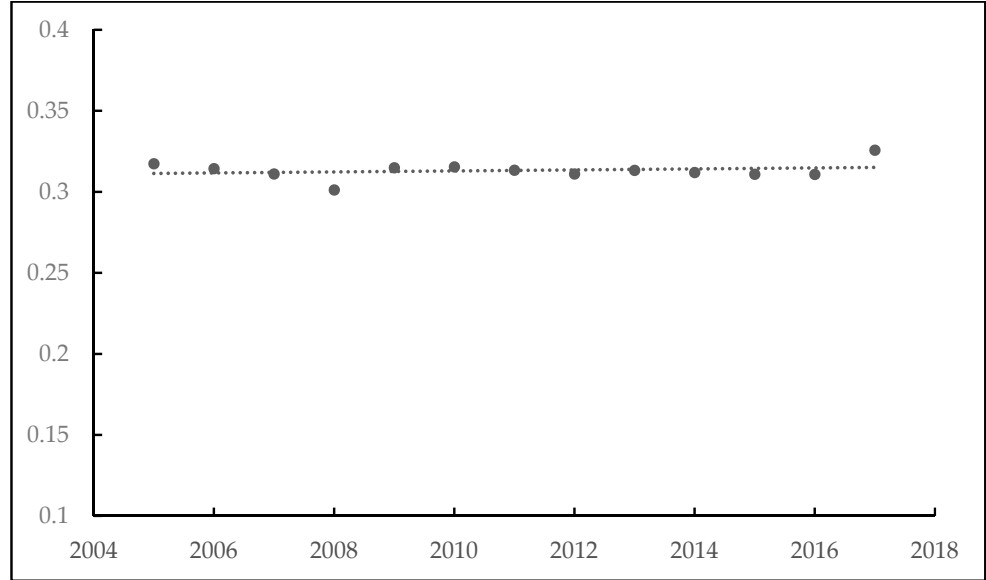

**Figure 3.** Change in the average value of the spatial structure of industry in the CPUA of China.



*4.2. Baseline Regression*

4.2.1. Parallel Trend Surface Test

First, we examine the parallel trend line using the averages of monocentric and polycentric spatial structure and industrial agglomeration of two groups. Figure 4 shows the spatial structure from the two perspectives of population and industry, which indicates that two groups have basically the identical trends before the opening of HSR. Additionally, we use more accurate regression statistics to assess the parallel trend hypothesis using formula (4). As shown in columns (1) and (2) of Tables 1 and 2, before HSR operation, the interaction coefficients of the year dummy variables and experimental group are not significant, which shows the parallel trend assumption is satisfied.

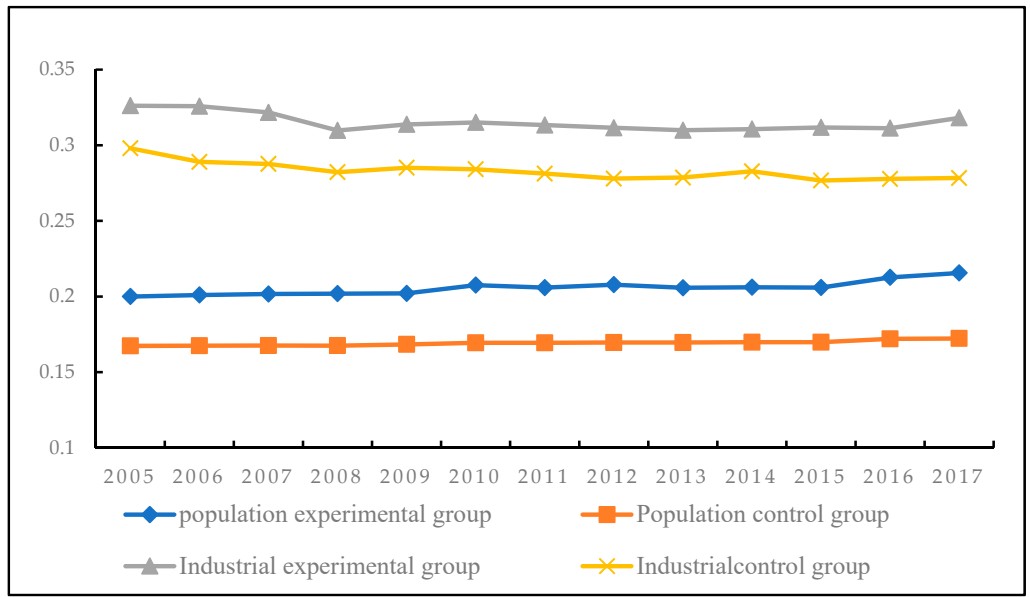

**Figure 4.** Parallel trend test.

**Table 2.** Baseline regression results for the spatial structure of population.

| Variable | (1) | (2) | (3) | (4) | (5) | (6) |
|---|---|---|---|---|---|---|
| treat*Year2007 | 0.068 | 0.014 | | | | |
| treat*Year2008 | 0.013 | 0.009 | | | | |
| treat*Year2009 | −0.059 | 0.007 | | | | |
| Time*city | 0.023 | | 0.031 | 0.031 | | |
| Number of trains | | −0.001 | | | 0.007 | 0.006 |
| adm | −0.123 * | −0.146 * | | −0.127 * | | −0.109 * |
| lnurbp | −0.243 *** | −0.412 *** | | −0.25 *** | | −0.351 *** |
| lnurbp$^2$ | 0.016 *** | 0.016 *** | | 0.016 *** | | 0.016 *** |
| lnpeGDP | 0.002 ** | 0.004 ** | | 0.006 ** | | 0.007 ** |
| lnpeGDP$^2$ | 0.061 ** | 0.050 ** | | 0.102 ** | | 0.151 ** |
| p sec-ter ind | 0.004 | 0.006 | | 0.001 | | 0.007 |
| p gov | −0.067 | −0.054 | | −0.043 | | −0.018 |
| cou | −0.016 *** | −0.015 *** | | −0.017 *** | | −0.017 *** |
| Year fixed effect | Yes | Yes | Yes | Yes | Yes | Yes |
| City fixed effect | Yes | Yes | Yes | Yes | Yes | Yes |
| R$^2$ | 0.3356 | 0.421 | 0.357 | 0.469 | 0.364 | 0.471 |

*, **, *** express significance at 10%, 5%, 1%.

4.2.2. Baseline Regression of the Spatial Structure of the Population

According to the regression results of columns (2) through (6) of Table 2, the effects of the variables of the implementation of HSR and the service level coefficient of HSR are not

statistically significant. Regardless of whether the control variables are included, there is no evidence that HSR and service strength are related to the monocentric and polycentric spatial structures of the population.

The coefficient of the administrative area is significantly negative, indicating that the larger the land area of the municipal administrative area is, the greater the possibility of regional spatial expansion and the more balanced the multicenter development of the spatial structure of the population. Every 1% increase in administrative area reduces the spatial structure value by 0.109. The first-order coefficient of the urban population is negatively significant, and the second-order coefficient is positively significant, indicating that there may be a "U"-shaped nonlinear relationship between urban population size and spatial structure. When the level of urbanization is relatively low, the urban spatial structure is monocentric, but with the development of the city, the urban population increases, and the spatial structure of the city area develops in a polycentric direction. This observation reflects the dynamic competition and balance of centripetal and centrifugal forces in the development of population spatial structure. The coefficients of the first and second terms of per capita GDP are both significantly positive, indicating that the higher the level of economic development is, the more monocentric the urban spatial structure of the population. For every 1% increase in GDP per capita, the spatial structure value of the population increases by 0.151. The county-level unit coefficient is significantly negative, indicating that the greater the number of county-level units in a city-level unit, the greater the level of competition among governments at the city level and the greater the tendency toward a polycentric spatial structure of population. The proportion of secondary and tertiary industries and government expenditure in GDP have no statistical significance; that is, there is no evidence that these two factors are related to the spatial structure of the urban population.

### 4.2.3. Baseline Regression of the Spatial Structure of Industry

According to the regression results in columns (3) and (4) of Table 3, whether the model contains control variables or not, the coefficients of the interaction terms of the implementation effect of HSR are significantly positive at the 5% level, indicating that HSR operation increases the degree of industrial agglomeration by 0.031. This result shows that HSR strengthens the development of the existing center of the city; thus, the urban industrial spatial structure tends to be polarized. According to the regression results in columns (5) and (6) of Table 2, in the absence of a control variable, the regression coefficient of the frequency of HSR trains is significantly positive. This result is more reliable when the control variable is added and illustrates that the HSR service level has a significant agglomeration effect on the urban spatial structure. For every 1% increase in HSR trains, the Gini coefficient of the spatial structure of industry increases by 0.006.

The coefficient of the administrative area is significantly negative, and this result is consistent with the spatial structure of the population. The coefficients of the first and second terms of GDP per capita are positive, indicating that as the economic development level improves, the industrial spatial structure in prefecture-level cities tends to develop in the direction of agglomeration. When controlling for only socioeconomic factors, the higher the economic development level is, the more the spatial structure of the city tends to be a single center. Currently, the dominant spatial structure of industry is mainly agglomeration. The proportion of secondary and tertiary industry is significantly positive, indicating that the higher the proportion is, the more concentrated the spatial structure of industry is. In terms of per capita GDP and the secondary and tertiary industry coefficients, the economic development level of most prefecture-level cities in the CPUA is currently in the agglomeration stage. In many prefecture-level cities, the talents, funds, and policy preferences of the counties and county-level cities under their jurisdiction are attracted by or transferred to central cities. The coefficient of the proportion of government expenditure is significantly positive, indicating that the greater the degree of government intervention in the market, the more concentrated the spatial structure of industry. For every 1% increase

in government expenditure, the spatial Gini coefficient increases by 0.038. The coefficient of the county-level unit is significantly negative, consistent with the spatial structure of the population. No evidence of a relationship between the urban population and the spatial structure of industry was found.

**Table 3.** Baseline regression results for the spatial structure of industry.

| Variable | (1) | (2) | (3) | (4) | (5) | (6) |
|---|---|---|---|---|---|---|
| treat*Year2007 | 0.041 | 0.060 | | | | |
| treat*Year2008 | 0.083 | 0.059 | | | | |
| treat*Year2009 | 0.063 | 0.084 | | | | |
| Time*city | 0.031 ** | | 0.021 ** | 0.031 ** | | |
| Number of trains | | −0.006 ** | | | 0.007 ** | 0.006 ** |
| adm | −0.223 *** | −0.172 * | | −0.226 *** | | −0.172 * |
| lnurbp | −0.146 | −0.129 | | −0.018 | | −0.121 |
| lnurbp$^2$ | 0.09 | 0.116 | | 0.098 | | 0.115 |
| lnpeGDP | 0.137 ** | 0.004 ** | | 0.137 ** | | 0.007 ** |
| lnpeGDP$^2$ | 0.001 * | 0.002 ** | | 0.001 ** | | 0.004 ** |
| p sec-ter ind | 0.017 *** | 0.012 ** | | 0.018 *** | | 0.015 *** |
| p gov | 0.017 *** | 0.024 *** | | 0.019 *** | | 0.038 *** |
| cou | −0.086 ** | −0.065 *** | | −0.085 *** | | −0.091 *** |
| Year fixed effect | Yes | Yes | Yes | Yes | Yes | Yes |
| City fixed effect | Yes | Yes | Yes | Yes | Yes | Yes |
| R$^2$ | 0.362 | 0.478 | 0.367 | 0.416 | 0.444 | 0.542 |

*, **, *** express significance at 10%, 5%, 1%.

### 4.3. Robustness Test

#### 4.3.1. Robustness Test of the Spatial Structure of the Population

According to the benchmark regression results, HSR has no significant impact on the monocentric and polycentric spatial structure of the population. To ascertain the robustness of the analysis results, we further use our replacement measurement index. First, the population spatial structure is measured through the Primacy index. Next, we evaluate whether the variables before and after HSR conform to the parallel trend and whether the mean differs between the experimental and control groups. Columns (1) and (2) of Table 3 show that before the HSR operation, the interaction coefficients of the year dummy variables and the experimental group were not statistically significant. Prior to the HSR operation, the spatial structure of the experimental and control groups had the same flat trend, so the parallel trend assumption was satisfied.

The regression results of the Primacy index are shown in Table 4. The interaction coefficients of HSR and the time dummy variables are not significant. In addition, HSR train frequency and the spatial structure coefficient are not significant. Consistent with the previous benchmark regression structure, the test results are robust. Because the population spatial structure index calculated by the Primacy index is basically consistent with the regression results of the Herfindahl index measures, and the sign and significance of the coefficients of each control variable basically remained unchanged, the conclusion of the benchmark regression is relatively stable.

#### 4.3.2. Robustness Test of the Spatial Structure of Industry
Placebo Test

When using the DID method to detect the problem of missing variables with policy effects, some indicators that are not observed may have key impacts on the results. To solve this problem, we used the 2005–2009 data of unopened HSR for a placebo test. The actual operation time of the HSR was 2010–2017. We speculate that the year in which HSR operates does not matter; that is, the HSR opening effect appears in any year. We artificially set the year as that when HSR opened. To test the effect, we artificially set the year to 5 years earlier than the actual time when the HSR opened. In this case, if HSR and its

service level still have a significant impact on the spatial structure of industry, then there is a certain deviation in the benchmark regression results. Otherwise, we consider the benchmark regression results to be robust.

**Table 4.** Robustness test results for the spatial structure of population.

| Variable | (1) | (2) | (3) | (4) | (5) | (6) |
|---|---|---|---|---|---|---|
| treat*Year2007 | 0.056 | 0.064 | | | | |
| treat*Year2008 | 0.053 | 0.019 | | | | |
| treat*Year2009 | −0.029 | 0.037 | | | | |
| Time*city | 0.016 | | 0.131 | 0.131 | | |
| Number of trains | | 0.003 | | | 0.001 | 0.003 |
| adm | −0.033 * | −0.106 * | | −0.037 * | | −0.109 * |
| lnurbp | −0.304 *** | −0.341 *** | | −0.321 *** | | −0.351 *** |
| lnurbp$^2$ | 0.116 *** | 0.016 *** | | 0.141 *** | | 0.016 *** |
| lnpeGDP | −0.020 ** | −0.016 ** | | −0.021 ** | | −0.017 ** |
| lnpeGDP$^2$ | 0.009 * | 0.007 ** | | 0.007 ** | | 0.151 ** |
| p sec-ter ind | 0.005 | 0.004 | | 0.005 | | 0.006 |
| p gov | −0.046 | −0.064 | | −0.054 | | −0.056 |
| County-level unit | −0.017 *** | −0.005 *** | | −0.017 *** | | −0.005 *** |
| Year fixed effect | Yes | Yes | Yes | Yes | Yes | Yes |
| City fixed effect | Yes | Yes | Yes | Yes | Yes | Yes |
| $R^2$ | 0.681 | 0.421 | 0.540 | 0.716 | 0.364 | 0.471 |

*, **, *** express significance at 10%, 5%, 1%.

The placebo test results are shown in Table 5. The coefficients of both HSR and its service level are not statistically significant, proving that when the HSR opening time is artificially set to 5 years prior, the treatment effect is not statistically significant. Therefore, we can conclude that the problem of missing variables is not significant; the hypothesis is not valid, and the placebo test is passed, which confirms that HSR and its service level have a policy effect on the spatial structure of industry and that the benchmark regression is robust.

**Table 5.** Placebo test results for the spatial structure of industry.

| Variable | (3) | (4) | (5) | (6) |
|---|---|---|---|---|
| Time*city | 0.104 | 0.091 | | |
| Number of trains | | | 0.005 | 0.005 |
| adm | | −0.166 *** | | −0.152 * |
| lnurbp | | −0.021 *** | | −0.032 *** |
| lnurbp$^2$ | | 0.025 *** | | 0.015 *** |
| lnpeGDP | | −0.107 ** | | −0.009 ** |
| lnpeGDP$^2$ | | 0.002 ** | | 0.006 ** |
| p sec-ter ind | | 0.047 *** | | 0.039 *** |
| p gov | | 0.039 *** | | 0.045 *** |
| cou | | −0.053 *** | | −0.065 *** |
| Year fixed effect | Yes | Yes | Yes | Yes |
| City fixed effect | Yes | Yes | Yes | Yes |
| $R^2$ | 0.467 | 0.516 | 0.344 | 0.537 |

*, **, *** express significance at 10%, 5%, 1%. Replacing the measurement index.

To ascertain the robustness of the analysis results and solve the measurement error, we further use a substitute explanatory variable. First, we measure the spatial structure of industry using the Herfindahl index and then check whether the distribution of the two sets of variables before and after the opening of HSR conforms to the parallel trend (Table 6). Columns (1) and (2) of Table 6 show that before the HSR operation, the interaction coefficients of the year dummy variables and the experimental group were not statistically significant. This indicates that before HSR operation, the spatial structure of the two sets of

variables had the same flat trend, and therefore, the parallel trend assumption is satisfied. Table 5 shows the double-difference DID test results of the industry Herfindahl index. HSR and its service level are all significant at the level of 5% or 10%, and the sign and significance of the coefficients of each control variable remain basically unchanged, so the conclusion of the benchmark regression is relatively robust.

**Table 6.** Baseline regression results for the spatial structure of industry using the Herfindahl index.

| Variable | (1) | (2) | (3) | (4) | (5) | (6) |
|---|---|---|---|---|---|---|
| treat*Year2007 | 0.027 | 0.022 | | | | |
| treat*Year2008 | 0.083 | 0.045 | | | | |
| treat*Year2009 | 0.033 | 0.043 | | | | |
| Time*city | 0.064 ** | | 0.045 ** | 0.064 ** | | |
| Number of trains | | −0.004 ** | | | 0.004 ** | 0.004 ** |
| adm | −0.159 *** | −0.172 * | | −0.166 *** | | −0.154 * |
| lnurbp | −0.039 | −0.092 | | −0.038 | | −0.132 |
| lnurbp$^2$ | 0.091 | 0.006 | | 0.08 | | 0.008 |
| lnpeGDP | 0.157 ** | 0.102 ** | | 0.142 ** | | 0.137 ** |
| lnpeGDP$^2$ | 0.006 * | 0.007 ** | | 0.006 ** | | 0.007 ** |
| p sec-ter ind | 0.004 *** | 0.003 ** | | 0.043 *** | | 0.002 *** |
| p gov | 0.017 *** | 0.011 *** | | 0.019 *** | | 0.013 *** |
| County-level unit | −0.046 ** | −0.035 *** | | −0.052 *** | | −0.037 *** |
| Year fixed effect | Yes | Yes | Yes | Yes | Yes | Yes |
| City fixed effect | Yes | Yes | Yes | Yes | Yes | Yes |
| R$^2$ | 0.423 | 0.476 | 0.462 | 0.564 | 0.501 | 0.524 |

*, **, *** express significance at 10%, 5%, 1%.

## 5. Mechanism Analysis

Why do HSR and its service level have no significant impact on population spatial structure? First, in the short term, HSR construction brings great space-time compression, increases the possibility of travel, and increases short-term population flow. However, with respect to increasing the population and changing the spatial distribution of the population, the time horizon of the impact on long-term migration is relatively long. In other words, people may choose to commute every day via HSR rather than migrate to large cities. Second, the operation time of HSR in the CPUA is relatively short. Different conclusions might be reached at longer observation times. Third, the urban development stage may play a role. At present, the agglomeration power of the CPUA is greater than that of decentralization, but the population agglomeration brought by HSR is not sufficient to change the spatial structure of the city area.

In this paper, the mechanism by which HSR influences the urban industrial spatial structure is analyzed in terms of the following aspects. (1) Differences in HSR resources between cities and in the opening sequence result in an unbalanced space-time compression effect, which causes obvious spatial differences in the degree of accessibility improvement and, in turn, urban location conditions and factor flow. (2) HSR causes an adjustment of industrial structure. As the results in the paper show, the proportion of secondary and tertiary industry is significantly positive, indicating that the higher this proportion is, the greater the tendency of the industrial spatial structure to be a single center. First, the HSR network can improve regional transportation accessibility and expand market scope. The expansion of market scope impacts the regional division of labor, thereby enhancing regional specialization and industrial structure. Second, on the one hand, HSR reduces the proportion of the urban manufacturing industry, thereby promoting the upgrading of the manufacturing industry; on the other hand, it increases the proportion of the urban service industry, especially the high-end service industry [20,58]. Third, the HSR network improves a city's overall traffic conditions and promotes the development of tertiary industries such as business, catering and accommodation, transportation, real estate, and tourism services. As a result, tertiary industry continues to grow, and some of the original industries are

gradually reduced, eliminated, or transferred to other regions. Fujita et al. maintain that secondary industry tends to be decentralized, whereas tertiary industry prefers cities with greater market potential and tends to be centralized [59]. The higher the proportion of secondary and tertiary industries is, the greater the tendency of the spatial structure of the city to be a single center. At present, urban agglomeration is dominant in the CPUA, which promotes the uni-centricity of industrial spatial structure.

(3) HSR improves the level of economic development. From the above, we can see that the higher the level of economic development is, the more inclined the spatial structure is toward a single center. First, HSR brings increased consumption demand, rapid interaction of information, knowledge and other elements, optimized industrial structure, and greater urban economic development. Second, at present, the CPUA is in the stage of urbanization and rapid economic development, and the agglomeration mechanism plays a leading role. For the independent administrative units within the CPUA, HSR construction will cause economic growth and industrial agglomeration in HSR cities and central cities, and the optimization and adjustment of industrial structure may cause the loss of talent and industrial agglomeration in cities with poor economic development levels and investment environments. This will increase the gap between HSR and non-HSR cities, central cities and marginal cities and promote the development trend of the single-center industrial spatial structure in the city.

## 6. Conclusions and Discussion

### 6.1. Conclusions

HSR can guide the agglomeration of specific functions of the city, strengthen and catalyze the formation of comprehensive function centers, become a center of urban spatial growth, redistribute urban spatial resources, and promote the formation of a multicenter city structure. In this paper, traditional and continuous DID models are utilized to study the impact of HSR on the spatial structure of prefecture-level cities from the two perspectives of population and industry. The major findings are as follows:

(1) From 2005 to 2017, in terms of population and industry, the spatial structure of population in the prefecture-level cities in the CPUA showed a monocentric trend, whereas the spatial structure of industry in the prefecture-level cities showed an agglomeration trend. However, different prefecture-level cities showed different characteristics. The spatial structure of the population in Luohe, Shangqiu, and Xinyang showed a weak polycentric trend, whereas the industrial spatial structure of Zhoukou and Puyang showed a weak polycentric trend, but all belonged to the average decentralized polycentric structure.

(2) Examining the impact of HSR on spatial structure from the two perspectives of population and industry revealed that HSR and its service level have basically no impact on population spatial structure in prefecture-level cities but do impact industry spatial structure. In terms of industry, HSR and its service level have a significant positive impact on the spatial structure of the city. HSR increases the degree of industrial single-center structure by 0.031, and for every 1% increase in HSR trains, the Gini coefficient of the spatial structure of industry increases by 0.006. In addition, the evolution of the spatial structure in prefecture-level cities is affected by many natural factors, socioeconomic factors, policies, and transportation factors. Among the controlling factors, the administrative area, economic development level, urban population, and number of research units are the key factors impacting the spatial structure of the population in prefecture-level cities in the CPUA. The main factors impacting the spatial structure of industry are the administrative area, socioeconomic factors, urban population, industrial structure, government intervention, and number of district and county units.

*6.2. Discussion*

The present study of China's prefecture-level cities fills a gap in existing research in this area. The characteristics and rules of urban spatial structure vary depending on the space and show scale dependence. Previous studies have examined the scale chain of region, city, and HSR station area [47,48,51], and the impact of HSR on urban spatial structure at the prefecture level has been neglected. Therefore, research on the city at the prefecture level will help establish a more complete and systematic scale research chain (region, prefecture level, municipal districts, and HSR station area). Most research on urban agglomeration polycentricity examines the morphological dimension using population data, and a multidimensional research perspective is lacking. Therefore, the present study on the impact of HSR on urban spatial structure according to the two aspects of population and industry enriches the literature. Previous studies of HSR mostly focus on the effectiveness of HSR policy implementation. On this basis, in this paper traditional and continuous DID models are constructed, and the impact of HSR opening and service level on urban spatial structure are analyzed. In this paper, the impact of HSR opening and service level on urban spatial structure are analyzed. The conclusions reached here provide important insights into the relationship between transport networks and urban structure, thus promoting regional sustainable development.

In addition, research on the effect of HSR on urban would be strengthened by the availability of multisource data, such as "micro blog sign-in data", "night light data", and "mobile phone signaling", which could improve the detail and accuracy of future research. Future targeted studies could be aimed to evaluate the effects of HSR on the municipalities belonging to a province hosting an HSR, thus exploiting an even more disaggregated geographical level; moreover, it would be interesting to consider the relationship among HSR stations, industries, and cities.

**Author Contributions:** Writing—original draft preparation, X.W.; writing—review and editing, J.L. and W.Z. All authors have read and agreed to the published version of the manuscript.

**Funding:** This research was funded by the Henan University, grant number CX3050A0950051.

**Institutional Review Board Statement:** Not applicable.

**Informed Consent Statement:** Not applicable.

**Data Availability Statement:** Data available on request due to restrictions in privacy.

**Conflicts of Interest:** The authors declare no conflict of interest.

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
