# Peer review of "Impact of High-Speed Rail on Spatial Structure in Prefecture-Level Cities: Evidence from the Central Plains Urban Agglomeration, China"

_sustainability, doi:10.3390/su142316312_

Round 1
Reviewer 1 Report (New Reviewer)
This is a very well written paper with clear and valid methodology, research objectives and results. It has in practice reached publication status in my opinion. I only have two minor comments for authors to consider:
1. In their literature review, authors should also report figures (i.e. percentages, values etc.) from the past studies they already discuss so as to better quantify the impacts of HSR on population and industry spatial structures.
2. In section 3.2 authors should provide a Table describing basic statistics for the data they used in their study.
Author Response
Please see the attachment

Reviewer 2 Report (New Reviewer)
The paper is too wordy and readability is poor. The overall presentation of the manuscript should be improved. The novelty should be justified with a good comparative study of existing works. Figures and tables should be improved.
Author Response
Please see the attachment

Reviewer 3 Report (New Reviewer)
The paper is interesting and it can be a good contribution to the established line of research on the impact of HSR. English has to be improved before publication. For example: Introduction, 1st paragraph, line 4: "an important method of travel" should change to "means of travel" or "travel mode". 3rd paragraph: "Spatial unequal" should change to "Spatial inequality" etc. Please revise the whole document with a keen eye for detail.
Also, please check the detailed comments below to improve the paper further:
-In the abstract, mention where Central Plains Urban Agglomeration is located. It should not be assumed that all readers are familiar with the region.
-1. Introduction:
2nd paragraph: the term "Shinkansen" is introduced without any explanation that it refers to the Japan high-speed train network. Again, do not assume that every reader is familiar to region-specific terms.
3rd paragraph: explain which cities are considered core and which marginal. Also the last part of this paragraph presents many arguments but without using any literature to support them, for instance: "The situation goes against the sustainable development of the city and furthermore, imposes restrictions on the sustainable development of social economy of the country and the regions." Is that a view of the authors, or there is published work to support this?
Last paragraph: the acronym DID models is introduced without explaining what it stands for. Also, the term "prefecture-level cities" needs to be explained.
-2. Literature review:
2nd paragraph: explain the term "urban belts".
-3. Settings, data, and method:
3.1: You mention: "The construction of a “meter” HSR network with Zhengzhou as the core by 2020 is planned." This is a bit outdated - 2020 is over now, so what happened? Was it realized as planned?
3.3: "difference-in-difference" (DID): now what the acronym means is explained, but still, why the reader should know what difference-in-difference method entails? You need to be more precise.
- 6. Conclusions and discussion
6.1. Conclusions: 1st paragraph: In the conclusion section of the paper, mainly discussion on your own research and results is expected, so mentioning cities in Spain to begin with seems a bit odd.
Author Response
Please see the attachment

Reviewer 4 Report (New Reviewer)
This paper studies the influence of HSR on the spatial structure in prefecture-level cities at the single/multi-center levels . The study is important and interesting. The following is my comments:
1.The authors should avoid using extreme words/expression, since they sound opinionated.
2. Does OD passenger flow demand influence the study?
3. It is suggested to add some predictions for future development.
Author Response
Please see the attachment

This manuscript is a resubmission of an earlier submission. The following is a list of the peer review reports and author responses from that submission.
Round 1
Reviewer 1 Report
The paper investigated the impact of high-speed rail (HSR) on the spatial structure in prefecture-level cities using the traditional difference-in-difference (DID) models. Authors argued that HSR and its service level have no impact on the spatial population structure in prefecture-level cities but impact the spatial industry structure. However, the critical challenges stated in the paper are not as straightforward as follows.
- Even though the DID model is intended to mitigate the effects of extraneous factors and selection bias, depending on how the treatment group is chosen, this method may still be subject to certain biases (e.g., mean regression, reverse causality and omitted variable bias). Thus, recent studies proposed improved methods (e.g., using Monte Carlo simulations). It is necessary to present how the current experiment overcomes these limitations.
- R^2 values are relatively low. Although different research areas have different levels, is it meaningful values in this research? Could you please provide alternative values such as effect size?
- The manuscript requires extensive editing of the English language. For example
Abstract:
multi center dimension, from the dimensions of ~ -> check ,
The result show that => check show
Page 3, Secon paragraph:
when HSR enters the urban systems => check when
Reviewer 2 Report
This paper is definitely up – to - date and interesting for readers and the quality of article is good. However, some supplementations and improvements are required. That is why, I would like to list of remarks or questions to the author(s).
General remarks:
- The aim of this manuscript should be clearly stated and emphasised in the Introduction.
- The research procedure with a use of scheme should be supplemented, now it is difficult to follow the author(s) without knowing individual stages of the study (please, supplement section 3).
- The ‘Mechanism analysis’ appears far too late in this article, a part of it should belong to the ‘Settings, data, and method’ giving a proper explanation before results and the other part should supplement the Discussion. I suggest reorganisation.
- The discussion is practically non-existent. It is recommended to differentiate it in a separate section.
Details:
- At the end of the introduction, I suggest adding a short description of the paper's organization.
- The CPUA acronym’s meaning was not defined in the first line, that appears - avoid using acronyms in the abstract if they are not defined previously. Besides, once it is CPUA and next time CPUR. Both should be defined.
- Some references are from the period of 1950 – 2000, I suggest updating.
Reviewer 3 Report
On the whole, the manuscript entitled "Impact of high-speed rail on Spatial Structure in Prefecture level Cities: Evidence from the Central Plains Urban Agglomeration, China" challenges an important topic of transport development and its link to broader spatial changes. The manuscript is well written as well structured which proves that its authors are well aware of the studied problematics and devoted to contributing to the current academical debate. However, there are some essential flaws that need to be solved in order to reconsider this manuscript to be published.
Even though it might seem that there is a lot to be improved, I encourage authors for doing so, because they have already succeeded in designing important research. The suggestions I have made will only improve the clarity of the article and its structure which will be beneficial for the readers.
At the broad level, I would recommend authors to place the article more explicitly within the current debate over the sustainable development/ sustainable mobility development. As the authors want this manuscript to be submitted in Sustainability journal, the issue of sustainability has to be more clearly emphasized.
What would additionally improve the soundness of the article is to mention in the Introduction and Literature review also the negative issues which are connected with the HSR development, e.g. brain-drain or social-just policies to be sure every member of our society could use the HSR. Especially the literature review is focusing mainly on the advantages. I consider this as a serious flaw.
I also recommend authors to clearly define at the end of Introduction the main aim and research questions as the authors are referring to this terminology later in the manuscript, but it is not explained elsewhere. E.g. in the last paragraph of the page 14 is explained which kind of aspects are analysed in the paper. This should appear earlier in the manuscript – preferably in the introduction.
The Introduction should contains at the very end information about how the manuscript is structured.
The section Mechanisms analyses should be written differently. Partly it seems like introduction explaining what is going to be studies and secondly, there are fragments which would be rather suitable for the discussion part. I recommend authors to think about re-difining this part of the manuscript.
Lastly, more work has to be done at the Discussion and Conclusion section. It is up to the authors, whether to explicitly write short one or two paragraph subsection devoted only to main conclusions and afterward the discussion subsection or to write the only Discussion. The Conclusion should cleary answer the question asked in the Introduction. However, what I consider as a flaw is absence of references and placing the study conclusion into the current academical debate. Are the results supporting the current stream of knowledge? How they differs? Those are an example of questions authrs should answer in the Discussion. Also, policy recommendation or direction of further research could be signalized in the Discussion part.
Recommendations:
- In the introduction, I recommend authors to explain the HSR shortcut at the place, where it appears first.
- Mention the figure 1 in the 3.1 earlier in the text. Preferably at the place where authors are describing the study area.
- Instead of Figure 1 it would be better to have two separates figure with different titles.
Reviewer 4 Report
The article utilizes traditional and continuous difference-in-differences (DID) models to explore the influence of the HSR on the spatial structure in prefecture-level cities from the single/multi center dimension. from the dimensions of polycentricity and dispersion. I suggest publishing the article in the current form.
Round 2
Reviewer 1 Report
Authors responded to all comments accordingly.
Reviewer 2 Report
Thank you for your careful revisions. Most of my comments from first round review were addressed. The paper has a significant improvement.
I have noticed a few minor editing errors. However, I think you will receive a clear guidance from the editorial office on how these errors should be improved.
Reviewer 3 Report
Firstly, I would like to congratulate the authors on the improvement of the manuscript. The readability and clarity, and logical coherence of the paper meet the paper's standards to be presented in the Sustainability journal.
I'm recommending the article to be accepted after minor revisions.
The minor revisions are as follows:
It is up to the authors, but I would rather see Conclusions followed by a Discussion chapter. I suggest to present the discussion in one section and to avoid subsections as the authors have provided (6.1, 6.2.)
What could be considered a serious flaw is that both of the Discussion subsections are full of statements such as: "previous work, most research, it is argued," etc., without providing references to concrete publications. This has to be sorted out.
The authors are also providing valuable information in section 5.1. and 5.3. I recommend the authors clearly indicate the sources of some of the findings or terms (e.g. Siphon effect,...)
Section 5.2., sentence: "As shown in previous work..." – at this place should be references to the previous work authors have mentioned.